# Dynamic nuclear magnetic resonance field sensing with part-per-trillion resolution

Simon Gross[1], Christoph Barmet[1,2], Benjamin E. Dietrich[1], David O. Brunner[1], Thomas Schmid[1] & Klaas P. Pruessmann[1]

High-field magnets of up to tens of teslas in strength advance applications in physics, chemistry and the life sciences. However, progress in generating such high fields has not been matched by corresponding advances in magnetic field measurement. Based mostly on nuclear magnetic resonance, dynamic high-field magnetometry is currently limited to resolutions in the nanotesla range. Here we report a concerted approach involving tailored materials, magnetostatics and detection electronics to enhance the resolution of nuclear magnetic resonance sensing by three orders of magnitude. The relative sensitivity thus achieved amounts to 1 part per trillion ($10^{-12}$). To exemplify this capability we demonstrate the direct detection and relaxometry of nuclear polarization and real-time recording of dynamic susceptibility effects related to human heart function. Enhanced high-field magnetometry will generally permit a fresh look at magnetic phenomena that scale with field strength. It also promises to facilitate the development and operation of high-field magnets.

[1] Institute for Biomedical Engineering, ETH Zurich and University of Zurich, Gloriastrasse 35, Zurich 8092, Switzerland. [2] Skope Magnetic Resonance Technologies AG, Gladbachstrasse 105, 8044 Zurich, Switzerland. Correspondence and requests for materials should be addressed to K.P.P. (email: pruessmann@biomed.ee.ethz.ch).

Recent years have seen sustained progress in the generation of ever stronger magnetic fields for a range of purposes including basic[1] and high-energy physics[2], nuclear magnetic resonance[3] (NMR) and biomedical imaging[4,5]. However, progress in generating such high fields has not been matched by corresponding advances in magnetic field measurement. Moderate magnetic fields up to several $100\,\mu T$ are measured most sensitively with superconducting quantum interference devices[6] (SQUID), atomic magnetometers[7], and nitrogen-vacancy techniques[8], reaching sensitivities of fractions of fT Hz$^{-1/2}$ (ref. 9). Yet, to-date these mechanisms are not feasible in high-field conditions. Among known measurement principles that are suitable for high-field operation, NMR magnetometry[10] outperforms other feasible options such as Hall plates or magnetoresistive devices, currently offering relative sensitivities of $5 \times 10^{-7}$ (ref. 11) to $10^{-9}$ (refs 12,13). More sensitive high-field magnetometry is required to reveal a range of smaller-scale field dynamics of interest. In particular, it would facilitate the observation of weak magnetic phenomena that scale with an external field such as magnetization dynamics of electronic and nuclear origin.

NMR magnetometry is based on measuring the frequency of nuclear spin precession, which is proportional to the local magnetic field. Today it is mostly performed in a pulsed fashion, that is, by acquiring a free induction decay (FID) signal after exciting a suitable sample with a radiofrequency pulse. To derive the magnetic field strength, the signal phase

$$\phi(t) \propto \int_0^t B(\tau)\mathrm{d}\tau \qquad (1)$$

is extracted, unwrapped and subject to temporal differentiation or regression. The measurement precision thus achieved scales as

$$\sigma_B \propto \frac{1}{\xi T_{\mathrm{obs}}^{3/2}} \qquad (2)$$

where $\xi = \mathrm{SNR}\sqrt{\mathrm{BW}}$ denotes the bandwidth-compensated signal-to-noise ratio (SNR, in amplitude terms) of the recorded NMR signal and $T_{\mathrm{obs}}$ the observation time[13,14].

For suitable sample materials the signal lifetime of NMR sensors is limited by inhomogeneous broadening due to field non-uniformity within the sample volume. Inhomogeneous broadening amounts to the superposition of signal components associated with different field strengths. At the level of signal phase such superposition is non-linear and hampers interpretation in terms of equation (1). Therefore the usable sensor lifetime $T_{\mathrm{obs}}$ is effectively limited to a fraction of the inverse net linewidth. Efforts to advance NMR magnetometry thus focus on minimizing field distortion within the sensor along with maximizing the signal strength, and high-SNR detection electronics.

Detrimental field inhomogeneity arises from all magnetized matter near the NMR sample and particularly in the probehead itself. Caused by magnetic susceptibility, this problem grows more severe as the field strength increases. For high-performing magnetometry it has been addressed by the use of gaseous samples at low pressure, a regime in which field non-uniformity is evened out by motional averaging[15]. However, the low density of such samples yields only weak signals unless countered by hyperpolarization[16], which takes substantial time and thus hampers time-resolved measurement.

In the following, we report enhanced NMR field sensing using liquid samples, which offer vastly higher spin density than gases. The use of such samples is enabled by addressing inhomogeneous broadening at its root by tailoring the magnetism and geometric configuration of the materials involved. In combination with advanced detection electronics this approach is shown to achieve relative field resolution of one part per trillion and absolute sensitivity in the picotesla range. To exemplify this capability we demonstrate the direct detection and relaxometry of nuclear polarization and real-time recording of dynamic susceptibility effects related to human heart function.

## Results

**The sensors.** The sensors used in this work are based on purified liquids with high density of magnetically equivalent $^1$H or $^{19}$F nuclei, such as $H_2O$ and $C_6F_6$, encapsulated in a borosilicate capillary. The capillary is surrounded by a copper solenoid for signal excitation and detection and encased in epoxy polymer (Fig. 1). To tailor the magnetostatics of this setup we first performed high-precision magnetic susceptibility measurement of the materials involved. This was achieved by spatial mapping of phase changes that a reference volume of target material induces in NMR signals of a surrounding liquid. The resulting phase maps were subject to regression based on corresponding maps of two calibration materials, water and air, yielding volume susceptibility values with precision and accuracy of a few parts per billion (p.p.b.). Based on such characterization the suscept-ibility of the epoxy material was then matched, again with p.p.b. precision, to that of the solenoid by doping the epoxy resin with $Cu^{2+}$ ions in solution. Uniform susceptibility outside the capil-lary, combined with ellipsoidal shape of the casing, prevents the magnetized probehead from distorting the field inside the capillary. Similarly, the sensor's epoxy neck was rendered magnetically invisible by doping with $Dy^{3+}$ such as to exactly match its susceptibility to the surrounding air. To prevent the formation of microscopic gaseous inclusions and variation in eventual volume susceptibility the epoxy resin was polymerized under temperature and volume control. MR field mapping confirmed the expected level of field uniformity in the sensitive volume at a s.d. of 5.7 p.p.b. Sensor signals were excited and acquired with custom-built instrumentation including a high-rate direct-undersampling spectrometer optimized for high-SNR operation (see Methods). Thermal relaxation of the detector liquids is controlled by doping with paramagnetic compounds, which permits minimizing dead times for high temporal resolu-tion. Notably, homogeneous broadening does not violate equation (1) and thus permits controlling temporal resolution without introducing systematic error. For different measurement purposes the temporal resolution was varied between 6 and 100 ms.

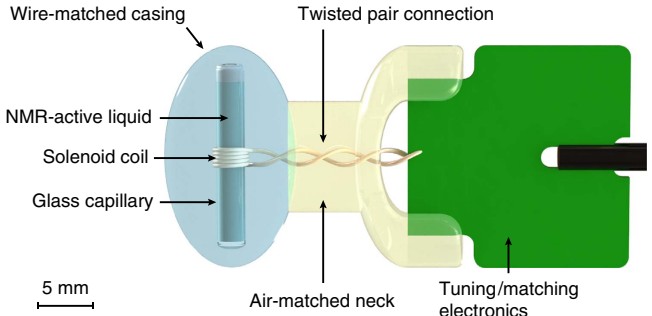

**Figure 1 | Sensor design.** The nuclear magnetic resonance (NMR) active sensor liquid is contained in a 2.2 mm diameter borosilicate capillary. A transmit/receive solenoid coil is tightly wound onto the capillary and encapsulated in an ellipsoidal epoxy casing. To prevent detrimental field non-uniformity the polymer is doped to exactly match the magnetic susceptibility of the coil wire. Likewise, the probe neck is susceptibility-matched to the surrounding air.

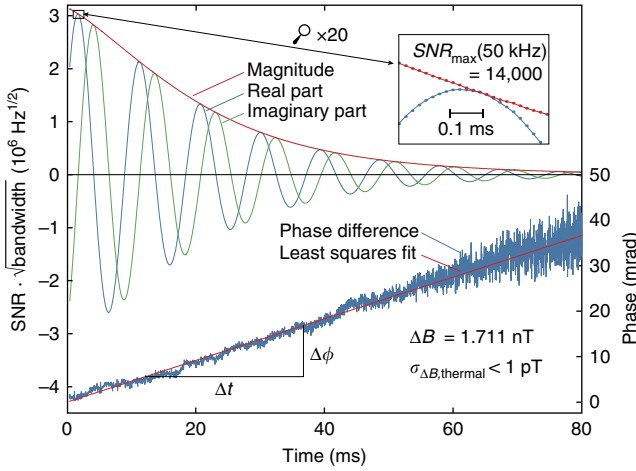

**Figure 2 | Measurement principle and raw signal of a $^1H_2O$ sensor operated at 7 tesla.** Primary signal-to-noise ratio (SNR, $3.1 \times 10^6$ Hz$^{1/2}$ in the data shown) and exploitable signal lifetime determine the sensor's intrinsic field sensitivity, which was assessed at better than 1 pT. The lower part of the figure shows the phase difference between a measurement and a calibration free induction decay (FID) signal (at 50 kHz bandwidth). The slope of the linear fit yields the change of magnetic field strength $\Delta B$ between the two measurements, a magnet drift in this case. Non-thermal noise in these data mostly reflects actual minute fluctuation of observed field.

Figure 2 illustrates the measurement principle and shows typical performance of a $CuSO_4$-doped $^1H_2O$ sensor laid out for operation in a background field of 7 T at a temporal resolution of 100 ms. The sensor's bandwidth-compensated initial SNR amounts to $\xi = 3.1 \times 10^6$ Hz$^{1/2}$ (Supplementary Fig. 1), corresponding to a phase noise level of $\sigma_\phi = 0.23$ µrad Hz$^{-1/2}$. On this basis, noise propagation, accounting for signal decay, yields a thermal sensitivity limit for the sensor itself of less than 1 pT ($\approx 0.14 \times 10^{-12}$ in relative terms). Overall sensitivity was governed by the receiver electronics whose phase noise limited the field precision to 6 pT and thus to below one part per trillion ($< 0.9 \times 10^{-12}$) for the given signal characteristics (see Methods and Supplementary Fig. 2 for details).

This level of sensitivity is realized in measurements of field dynamics as targeted in this work. For absolute measurements of static fields, raw field values would require correction for the field offset caused by the magnetized sensor material. This offset is readily calculated from the known susceptibilities and geometry[17,18]. However, the accuracy of resulting absolute field values will still be fundamentally limited by the remaining uncertainty of the gyromagnetic ratio of the proton, which currently amounts to 6.9 p.p.b.[19].

**Sensing of nuclear magnetism.** One immediate area of application of enhanced high-field magnetometry is nuclear magnetism itself. The transverse components of nuclear magnetization are subject to Larmor precession and hence readily observed by induction, typically in the high MHz range. By contrast, the axial magnetization component varies much more slowly, typically on the scale of milliseconds to seconds, and thus eludes inductive detection. Nuclear polarization is also exceedingly weak with the largest nuclear susceptibilities reaching the order of $10^{-9}$. In low background field, detection of such faint axial magnetization has been accomplished using SQUIDs[20], atomic magnetometers[21], anisotropic magnetoresistance[22] and nitrogen-vacancy centres in diamond[23]. However, in the high-field realm, where nuclear magnetism is most commonly studied and used, the ability

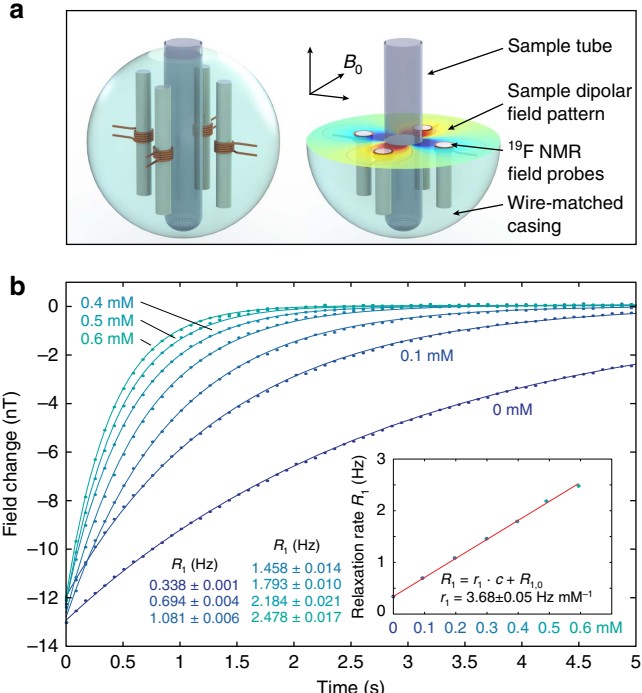

**Figure 3 | Direct observation of axial nuclear magnetization.**
(**a**) The sample substance is contained in a cylindrical glass vial at the centre of the setup placed in a 7-tesla magnet. The dipole field of its nuclear magnetization is sampled by four $^{19}$F nuclear magnetic resonance sensors (figure not exactly to scale, capillaries and distances magnified by a factor 2 for visibility). (**b**) Axial $^1$H relaxation in water at varying concentration of gadoteric acid. Regression of fitted relaxation rates yields a high-precision estimate of the dopant's relaxivity.

to observe axial nuclear magnetization is limited to-date. In magnetic resonance force microscopy[24], it is sensed via force coupling to a detection cantilever. Yet, relying on active nutation of the magnetization in question this mechanism perturbs and largely masks the native spin dynamics. Another option, the Hall effect, has been successfully used to detect axial electronic polarization[25] yet remains to be rendered sufficiently sensitive for the nuclear pendant.

At the level of sensitivity reported above, enhanced NMR sensors can readily fill this gap. When used to measure the magnetic field generated by other atomic nuclei they effectively leverage dipolar coupling, which occurs both between single spins[26–28] and remotely between spin ensembles[29–31]. Here we report the use of four $^{19}$F NMR sensors to measure field excursions produced by magnetization dynamics of $^1$H nuclei. Arranged in the fashion shown in Fig. 3a, their recordings were averaged with alternating sign such as to capture the dipolar field pattern of the $^1$H sample while suppressing external field fluctuations and clock jitter. As an example, Fig. 3b shows use of this setup for the direct observation of spin-lattice relaxation in $^1H_2O$. After pulsed spin inversion the ensuing recovery is recorded by continuous field measurement at a temporal resolution of 84 ms. An immediate application of this capability is relaxometry as illustrated by doping the water sample with varying concentrations of gadoteric acid, a magnetic resonance imaging (MRI) contrast agent. The obtained data were found to conform excellently to expected exponential behaviour, which validates the actual observation of axial nuclear magnetization dynamics. Based on single experiments of several seconds each, exponential fitting yielded the resulting relaxation rates with precisions better than 1%, comparing favourably with

conventional fast methods of spin-lattice relaxometry such as the Look-Locker technique[32,33]. An even more salient benefit of direct nuclear relaxometry is the avoidance of systematic errors that afflict NMR relaxometry with transverse detection. Residual error of radiofrequency transmission and confounding spin and higher-order echoes typically limit the accuracy of Look-Locker techniques, for instance, to several percent[33]. High accuracy of direct sensing is confirmed by the inset graph, which reflects the linear relationship between the dopant concentration and the measured relaxation rate. It immediately yields the dopant's relaxivity, $r_1$, which is a key quantity in contrast agent design and application.

For the broader field of NMR the capability of observing axial magnetization in a direct, time-resolved fashion is a novel, generic means of investigation. For instance, it enables studies also of more complex magnetization dynamics involving phenomena such as cross-relaxation, magnetization transfer and chemical exchange. An intriguing variant of such studies will target hyperpolarized samples whose enhanced level of polarization is expected to reveal yet finer detail of axial magnetization dynamics. Given the added effort of preparing suitable hyperpolarized states it is particularly beneficial that direct magnetometry does not diminish or otherwise alter axial magnetization under study. Further promising applications include nuclear magnetization studies of solid-state samples that hamper inductive detection by fast transverse relaxation.

**Background fluctuation.** In the reported measurement of axial nuclear magnetization the recorded relaxation curves deviated from exact exponentials by 26–72 pT (root-mean-square error) and thus by significantly more than the previously assessed level of sensitivity. The discrepancy arises from incomplete gradiometric cancellation of fluctuations of the ambient magnetic field rather than from error introduced by the sensors. This was confirmed by a stability measurement of the magnet used, a 7-tesla superconducting electromagnet designed for MRI in humans. We used two $^1H_2O$ sensors of the same type as above, placed close to each other at a distance of 1.2 cm, and measured the two field strengths simultaneously at a temporal resolution of 100 ms. The recordings exhibited s.d. of 306 pT (over 1 s) to

428 pT (over 10 s) and statistics of non-thermal nature (Supplementary Fig. 3). The difference of the two time series was found to fluctuate much less, with s.d. of 30 and 43 pT over periods of 1 and 10 s, respectively. This indicates that the fluctuating readouts mostly reflect spatially coherent fluctuation of the background field rather than detection noise. In the difference the noise spectrum was still not flat (Supplementary Fig. 3) as it would be for intrinsic sensor noise. It thus indicates that the fluctuation of the background field involves spatially varying components that differ up to several tens of pT between the two sensors. This is conceivable given a range of potential fluctuation sources such as mechanical behaviour of the superconducting magnet.

**Observation of the beating heart.** Besides nuclear magnetism, enhanced high-field magnetometry also expands the capability of observing magnetization of electronic nature. The magnetic field that emanates from magnetized material offers non-invasive access to a large variety of observables ranging from material properties to chemical, biological, and even physiological processes. As an example of the latter, we demonstrate the recording of dynamic susceptibility effects caused by the beating human heart when exposed to an external magnetic field. Termed magnetic susceptibility plethysmography (MSPG)[34], such measurement has so far been limited to SQUID detection and thus to low background field[34,35]. At field strengths below 1 mT, the method has yielded cardiac field SNR of about 20. To improve net sensitivity, MSPG signals have typically been averaged over large numbers of heartbeats, capturing only dominant features and no beat-to-beat variation[35].

Here we report MSPG in real-time and with vastly higher sensitivity, enabled by high ambient field and NMR detection. Three NMR sensors were placed on a healthy volunteer for operation in the 7 T magnet used previously. One was positioned on the sternum, one close to the apex of the heart above the fifth intercostal space and one on the neck near the right carotid artery. The sensors were operated simultaneously at a temporal resolution of 6 ms, yielding cardiac field SNRs in excess of 4,000. Figure 4 shows resulting real-time readouts along with a concurrently acquired electrocardiogram (ECG). The field dynamics primarily reflect mechanics of magnetized material such as myocardial contraction, distension of the aorta and valve closure. Likely they also involve magnetohydrodynamics arising from electric charges in flowing blood[36], which are equally visible in the ECG in Fig. 4. As proposed in the earlier MSPG literature, such data hold promise to permit extraction of physiological parameters and potentially of diagnostic information[34,35]. Real-time recordings of greatly enhanced dynamic range, as reported here, render a major boost to these prospects. Other, immediate applications include cardiac synchronization of MRI scans, which today depends on ECG and tends to be unreliable at high field, as well as the use of MSPG regressors for removing physiological confound from fMRI time series[37].

**Discussion**

These examples illustrate the utility of expanded measurement capability with the proposed sensor concept. In a background of 7 T, the achieved intrinsic sensitivity was assessed at 1 pT at a temporal resolution of 100 ms. Signal integrity and field precision hinge on rigorous susceptibility management and extremely high primary signal-to-noise ratios that pose a major challenge to receiver electronics. With the current custom-designed receiver instrumentation an effective sensitivity of 6 pT has been accomplished. Such resolution marks a new level of sensitivity for time-resolved magnetometry in high background fields. It is

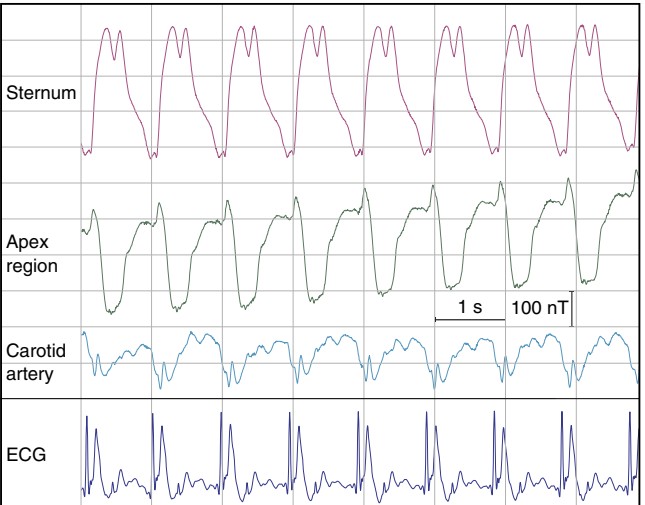

**Figure 4 | Real-time magnetic field recordings of cardiac activity.**
Magnetic field dynamics generated by the beating human heart in a background of 7 T, recorded at three different positions on the chest and neck, along with simultaneous electrocardiogram (ECG). Data were acquired during breathholding.

thus of immediate utility to expanding efforts in the design, construction and use of ever-higher-field magnets. Efforts to push sensitivity beyond the level now achieved are to focus on further improvement of receiver electronics. Added sensitivity margin at the receiver stage could then be leveraged by even longer signal acquisition from longer-$T_2$ samples. For large probe volumes, longer $T_2$ may eventually render so-called radiation damping[38] a limiting factor, which however could readily be addressed by established means such as a suitable feedback circuit[39].

Beyond magnet technology, detecting minute field changes in high backgrounds is particularly promising for the study and exploitation of magnetic phenomena that scale with field strength. Prominent examples are nuclear and electronic polarization and related dynamics as demonstrated here. In addition to boosting absolute sensitivity at high field, the reported approach also affords superior relative sensitivity. At the level of 1 part per trillion it is competitive with the most sensitive low-field magnetometry techniques devised to-date. Relative sensitivity of this order is even unprecedented at sub-second temporal resolution, enabling a fresh look at a range of dynamic magnetic phenomena such as those highlighted in this work.

## Methods

### Measurement and matching of magnetic susceptibility.
Susceptibility measurements were performed using a custom MRI setup, consisting of a spherical, water-filled phantom surrounded by a loop coil for inductive signal detection. At its centre the phantom included a cylindrical sample vial orthogonal to the static magnetic field, precision-mounted for exact repositionability. Spin-warp gradient-echo MR images were acquired in the central plane orthogonal to the vial. Long echo times up to 80 ms served to encode the static field distribution in the phantom, composed of a static background and field distortion proportional to the sample's susceptibility, in the resulting image phase. The setup was calibrated by imaging with the vial filled with two reference materials of known susceptibility, air and water. The susceptibility of each unknown third substance was then obtained by fitting a linear combination of the calibration maps to its imaged field footprint. Large statistical power of multiple 10,000 mapped positions (pixels), combined with pronounced field structure, render this method highly sensitive. Its precision was assessed at better than 5 p.p.b. (volume susceptibility, SI units). All measurements were performed at 3 T, using a whole-body human MRI system (Philips Healthcare, The Netherlands). Susceptibility matching of the epoxy polymers varied somewhat with lot number, at concentrations of close to 20 mM $Cu(II)(NO_3)_2$ (probehead ellipsoid) and 22 mM $Dy(III)(NO_3)_3$ (air-matched neck). Uniform solid solution in the cured material was achieved by dissolving the dopant in the epoxy resin, using acetone as a shuttle. A hard-curing novolac/nonyl-phenol epoxy system was chosen to minimize the lifetime of $^1H$ NMR signal from the polymer casts. Residual field non-uniformity was determined by means of a field mapping experiment carried out on a 7 T small animal MR scanner (Bruker Biospin, Ettlingen, Germany). A field map of the NMR-active probe volume was obtained by analysis of phase differences between 3D image data obtained with two different echo times.

### Electric properties of the sensor head.
The five-turn solenoid coil (2.9 mm diameter, 2.1 mm height, $L = 70$ nH) was wound from relatively thick (400 $\mu$m) high-purity Cu-ETP wire to contain ohmic losses. Its equivalent series resistance at 298 MHz was determined at 1.8 $\Omega$. The maximum electromotive force induced in the coil by the precessing nuclear magnetization of a $^1H_2O$ sample was determined at 700 $\mu$V (ref. 40). In comparison, the coil resistance translates into thermal noise of 0.17 nV Hz$^{-1/2}$ at room temperature, resulting in an upper SNR bound of $\xi = 4.1 \times 10^6$ Hz$^{1/2}$. The coil was tuned to the NMR frequency and matched to the pre-amplifier (50 $\Omega$) using a capacitive balanced matching network[41], employing only high-Q/low-loss, non-magnetic ceramic capacitors. We used a printed circuit board substrate optimized for the radiofrequency range. The tuned and matched sensor head has a Q-factor of 36. Tuning and matching imply impedance transformation and associated scaling of voltages. The latter affects signal and noise originating from the coil in the same way such that the SNR estimate remains valid up to losses in the added circuitry[40]. Figure 2 reveals the onset of mild radiation damping[38], which could be addressed by un-damping measures[39], yet was not considered limiting for the timing regimes targeted in this work.

### Radiofrequency spectrometer.
The spectrometer was laid out according to software-defined radio architecture, leveraging ongoing advances in high-speed digitization. The analogue part consists of a first, in-field low-noise amplification stage (low-noise monolithic microwave integrated circuit (MMIC), noise figure 0.6 dB, gain of 22.6 dB (500 MHz), $Z_{opt} = 50\,\Omega$) and a second, variable-gain ($-23.8$ dB to $+57.4$ dB gain) and filtering stage outside the magnet. The amplified

signals are digitized by direct undersampling with a high-speed analogue-to-digital-converter (250 MS s$^{-1}$, 14 bit). Quadrature demodulation, two-stage digital band-pass filtering and down-conversion are implemented on field-programmable gate arrays. Band-pass filtering entails an SNR increase in the time domain in proportion to BW$^{-1/2}$, which is supported by commensurate increases in bit depth along the digital processing pipeline. The output dynamic range is 32 bit each for the real and imaginary parts of the data. A precision oven-controlled crystal oscillator serves as a primary frequency standard and is used to generate the digitization clock.

### Field determination and sensitivity.
The bandwidth-corrected sensor SNR, $\xi$, is calculated as the ratio of the signal magnitude of the sensor's digital FID signal and the s.d. of the concomitant complex-valued noise, assessed through acquisition without NMR excitation, multiplied by the square-root of the signal bandwidth (Supplementary Fig. 1). The corresponding phase s.d. $\sigma_\phi$ is given by $\sigma_\phi / \sqrt{BW} = 1/(\sqrt{2}\xi)$. The field strength $B$ is obtained by linear regression of the unwrapped FID phase[13,14], discarding the initial 500 $\mu$s to allow complete decay of epoxy signal. The fitting duration was chosen between 5 ms and a maximum of 50 ms according to the targeted temporal resolution. The upper bound was still a small fraction of the inverse field s.d. to contain the influence of residual field non-uniformity. The thermal field precision $\sigma_B$ is obtained by calculating the noise covariance matrix of the regression results[13,14], accounting for signal decay caused by relaxation and dephasing. All field-sensor measurements were performed in a superconducting, passively shielded 7-tesla whole-body MRI magnet (Philips Healthcare, The Netherlands).

### Spectrometer stability.
To estimate the effect of spectrometer imperfection on field precision, we analysed a series of 1,000 FID signals (10 per second) from one sensor routed through two receiver channels. To perform the latter we used a 90° hybrid coupler splitting the signal before the first gain stage. This ensures that the two signals are identical up to a 90° phase shift and any thermal noise and perturbations that originate in and beyond the splitter. In particular, actual magnetic field fluctuation and clock jitter are identically reflected in both. A field sensor (rather than, for example, a synthesizer) was used as the signal source to exactly reflect the properties of actual sensor signals including their exponential decay. From the two digitized receiver outputs field time series were calculated (fitting duration 45 ms). Their difference exhibited a s.d. of 8.4 pT with an essentially flat power spectrum indicating phase alterations of predominantly thermal origin (Supplementary Fig. 2). Owing to the physical separation of the two receiver channels their phase noise is assumed to be uncorrelated, thus amounting to the equivalent of $8.4/\sqrt{2} = 6$ pT per channel. As expected from passive power splitting and the associated SNR loss of 3 dB, the thermal sensitivity limit of the difference data was assessed at $\approx 1.5$ pT and thus roughly $\sqrt{2}$ above the value of 1.0 pT obtained without splitting.

### Field stability measurement.
For precision field measurements all expendable auxiliary systems of the superconducting magnet were disabled to minimize field fluctuation. The setup was given time to settle into thermal equilibrium before the measurements were performed. Two $CuSO_4$-doped $^1H_2O$ field sensors were located in the isocentre of the magnet and separated (1.2 cm) by a removable, susceptibility-matched spacer, reducing mutual field distortions by the sensor heads. Slight radiofrequency crosstalk due to the close proximity of the two sensors was determined and eliminated in a post-processing step prior to regression for field determination. Field stability measurements were carried out in 11 runs of 1,000 field measurements each, with repetition times of 80, 100 and 200 ms. The fitting duration was 45 ms. The resulting field time series and their differences exhibited typical $1/f$ statistics. s.d. was therefore determined as a function of window length, obtained by averaging respective interval s.d. over the entire data sets of all runs (see Supplementary Fig. 3).

### Observation of axial nuclear magnetization.
For the acquisition of axial nuclear magnetization data (Fig. 3b), the setup illustrated in Fig. 3a was placed in a $^1H$ volume resonator (Nova Medical, Wilmington, MA, USA), which served for the application of frequency-modulated $^1H$ inversion pulses. The sample was contained in a 10.5 mm diameter glass vial. The $^{19}F$ field sensors contained CrTMHD$_3$-doped hexafluorobenzene ($C_6F_6$) in 2.2 mm inner-diameter borosilicate capillaries. The inversion pulse was followed by continuous field measurement with temporal resolution of 84 ms and a fitting duration of 50 ms. The field time courses of the four sensors were averaged with alternating sign, cancelling field fluctuations of zeroth and first spatial order as well as clock jitter. Contaminating NMR signal from $^1H$ contained in the setup was determined by reference experiments with $D_2O$ samples, ensuring equivalent loading and shimming conditions, and subtracted from eventual measurements. Mono-exponential fitting was performed over the entire data sets of up to 25 s using a nonlinear least-squares solver. The experiments were repeated 20 times, yielding average 95%-confidence intervals for the fitted relaxation rates as given in the inset table in Fig. 3b and root-mean-square errors of $26 - 72$ pT (see Supplementary Fig. 4). Fitting of the contrast agent relaxivity $r_1$ was performed using total least squares, accounting for estimated concentration uncertainties. The $R^2$ of the fit was 0.9988 with the maximum deviation amounting to 1.8%. Experiments were conducted at room temperature.

**Cardiac signatures**. MSPG data were recorded with GdCl₃-doped ¹H₂O, 2.2 mm inside-diameter sensors at a temporal resolution of 6 ms (fitting duration 5 ms). The thermal sensitivity limit of these fast sensors was determined at $\sigma_B = 65$ pT. The data shown in Fig. 4 were acquired with the probeheads placed on the chest and neck of a healthy volunteer at rest holding his breath. ECG data were acquired concurrently, using equipment built into the MRI system and synchronized with the field data by means of a reference gradient pulse.

**Data availability**. The data that support the findings of this work are available from the corresponding author on request.

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

## Acknowledgements

The authors are grateful to Stephen Wheeler for advice on and producing custom mechanical parts. Technical support from Philips Healthcare, particularly from Urs Sturzenegger, is gratefully acknowledged.

## Author contributions

S.G.: concept, sensor design and construction, susceptometry, experimental design, experiments, data analysis, manuscript. C.B.: conceptual considerations, sensor design, susceptometry. B.E.D.: spectrometer electronics and programming. D.O.B.: conceptual considerations, spectrometer electronics. T.S.: sensor design. K.P.P.: initiation, sensor and application concepts, data review, supervision, manuscript.

## Additional information

**Competing financial interests:** K.P.P. holds a research agreement and receives research support from Philips Healthcare. He is a shareholder of Gyrotools LLC and Skope Magnetic Resonance Technologies Inc. C.B. and D.O.B. are associates and shareholders of Skope Magnetic Resonance Technologies Inc. All other authors declare no competing financial interests.

