## [Peer Review File · Nature Communications]

(A) Reviewers' comments:

Reviewer #1 (Remarks to the Author):

This is a revised manuscript describing instrumentation and methods for very precise high-field magnetometry.

On the whole, the revisions address most of the concerns raised by the prior review. It is a little puzzling, however, that the manuscript still does not explicitly describe the magnetic field uniformity within the probehead, as assessed by the FWHM of the frequency spectrum. The FWHM was stated as 4.5Hz (15ppb) in the response letter, with the caveat that radiation damping may also contribute to the linewidth, but this detail is apparently absent from the manuscript itself.

The B_0 homogeneity is important for two reasons. First, it is a standard figure of merit for describing the performance of a susceptibility-matched probehead. Second, as noted in lines 39-41 of the manuscript, inhomogeneous broadening resulting from B_0 variations is a limitation of the technique. For the stated 4.5Hz linewidth, the T_2^* is approximately 70ms, implying that the method should start to break down for readout times that are a fraction of 70ms. In Fig. 2 (and, apparently, elsewhere), data are acquired and fitted over a readout time of 80ms. It is apparent from the FID in Fig. 2 that homogeneous broadening dominates inhomogeneous broadening, and moreover the authors note in their response letter that radiation damping may also play a role. However, there is still some lack of clarity with respect to the role of residual B_0 homogeneity that remains after the very careful susceptibility matching.

The manuscript should explicitly state the T_2^* or linewidth obtained with the probehead (with applicable caveats) and describe how this impacts the sensitivity of the technique.

Lastly, in supplementary Figure 2, "to closely spaced sensors" should be "two closely spaced sensors."

Reviewer #2 (Remarks to the Author):

The authors have carefully taken into account the input of all three referees and have substantially modified and improved the paper. The changes are also extensively described in the Rebuttal letter.

I think the paper should now be accepted for publication in its current form.

Reviewer #3 (Remarks to the Author):

In the revised version the authors have addressed some of my concerns and improved the readability of the manuscript.

Unfortunately the authors have left open a key question with respect to the true signal to noise ratio, which according to unnumbered equation (2) in line 34 of the manuscript determines the magnetic field measurement precision.

The claimed SNR = 14000 @ 50 kHz bandwidth is very suspect for the following reasons:

1. Thanks to the authors clarifications the electrical parameters and the operational conditions for the input circuit are now known. Given the inductivity of $L = 70$ nH, quality factor $Q = 36$ @ $\omega = 2\pi \cdot 500$ MHz we can calculate the AC resistance R of the coil, which is $R = \omega L/Q = 6.1$ Ohm. The impedance of the tuned (but not matched) circuit would be $R_t = R Q^2 = 7.9$ kOhm, which without matching results in a Johnson noise of the purely tuned circuit of about $N_t \sim 11.4$ nV/Hz^(-1/2). This of course is not the noise seen at the input of the preamplifier. The tuned and matched input circuit has an impedance of 50 Ohm, which correspond to a Johnson noise of $N \sim 0.91$ nV/Hz^(-1/2). At 50 kHz measurement bandwidth this results in a minimum noise of $N(50 \text{ kHz}) = 0.9$ nV/Hz^(-1/2) $(50000 \text{ Hz})^{(1/2)} = 204$ nV ~ 0.2 μ V. The authors corrected my estimated NMR signal amplitude from EMF ~ 200 μ V to 700 μ V, which I believe now to be true. This means that the best possible SNR achievable at the input coil in the presented experiment is $SNR_{max} = 700 \mu\text{V}/0.2 \mu\text{V} = 3500$, and not 14000 as claimed. Even a perfect preamplifier with a noise figure of 0 dB would not improve SNR_{max} beyond this ultimate limit, and neither will do any postprocessing, digital filtering etc. for a single scan.

2. In reality, there will always be a noise contribution from the preamplifier (PMA 5452 with unknown input current and input voltage noise) and additional noise sources which in practice can only diminish the SNR_{max} = 3500 observed. The only way for a fair measurement of the SNR which is essential for the parts-per-trillion claim of the manuscript is to show the raw data of the FID and the corresponding base line fluctuations (expanded appropriately). Fig.2 should be shown in recorded Volts over time and SNR could be determined by taking the maximum amplitude (e.g. $V_{peakpeak}/2$) dividing by the RMS voltage value of the noise fluctuations. The authors themselves admit that the baseline of the measurement “would however merely show a common stochastic noise trace”, but their claimed baseline for a SNR=14000 and 14 bit digital resolution cannot be a stochastic noise trace for it would only span the range of the least significant digital bit. For these two reasons, it is mandatory to show the reader the true baseline noise and the true SNR obtained in the experiments.

I think the authors should not destroy their remarkable results (Figs. 3, 4) by claiming a SNR which is not consistent with basic physics. Even if the claimed SNR was only a mere 1400, i.e. a magnitude lower than what was claimed, the resolution of the instrument would still be an impressive 10 parts-per-trillion and could be published in NatComm.

Reviewer #1

This is a revised manuscript describing instrumentation and methods for very precise high-field magnetometry.

On the whole, the revisions address most of the concerns raised by the prior review. It is a little puzzling, however, that the manuscript still does not explicitly describe the magnetic field uniformity within the probehead, as assessed by the FWHM of the frequency spectrum. The FWHM was stated as 4.5Hz (15ppb) in the response letter, with the caveat that radiation damping may also contribute to the linewidth, but this detail is apparently absent from the manuscript itself.

The B0 homogeneity is important for two reasons. First, it is a standard figure of merit for describing the performance of a susceptibility-matched probehead. Second, as noted in lines 39-41 of the manuscript, inhomogeneous broadening resulting from B0 variations is a limitation of the technique. For the stated 4.5Hz linewidth, the T2* is approximately 70ms, implying that the method should start to break down for readout times that are a fraction of 70ms. In Fig. 2 (and, apparently, elsewhere), data are acquired and fitted over a readout time of 80ms. It is apparent from the FID in Fig. 2 that homogeneous broadening dominates inhomogeneous broadening, and moreover the authors note in their response letter that radiation damping may also play a role. However, there is still some lack of clarity with respect to the role of residual B0 homogeneity that remains after the very careful susceptibility matching.

The manuscript should explicitly state the T2* or linewidth obtained with the probehead (with applicable caveats) and describe how this impacts the sensitivity of the technique.

Again, we appreciate the reviewer's efforts in examining our manuscript. The issue of field uniformity inside the sensor droplet is indeed critical and we are glad to address it in more depth.

In the previous response letter we mentioned the FID linewidth of 4.5 Hz as an upper bound of field variation. Regarding the success of susceptibility matching one must consider, however, that the FIDs are recorded in the presence of slight gradients in the background field that are unrelated to susceptibility in the sensor. Furthermore, the linewidth reflects not only inhomogeneous broadening related to static field non-uniformity but also homogeneous broadening due to T2 relaxation as well as potential radiation damping. We therefore concluded that quantification of the success of susceptibility matching, as the reviewer suggests, is best performed by mapping the static field in the droplet in addition to the linewidth estimate. We now report MR-based field mapping, yielding a field standard deviation 5.7 ppb, which is well in line with the estimated error margins of susceptometry (5 ppb) and susceptibility matching (few ppb). The FWHM of the corresponding frequency distribution is 2.1 Hz. Translating these values into T2* is not straightforward because it requires assumptions about lineshape. For a Lorentzian, the FWHM of 2.1 Hz would translate into a T2* of 150 ms. Yet since the notion of T2* does not quite apply to general frequency distributions we would prefer to state only the field uniformity, which directly reflects the success of susceptibility matching.

Regarding the effective signal lengths used in operating the sensors we realize that Fig. 2 may have caused confusion. The figure shows an extended length (80 ms) of a sample signal to illustrate increasing phase noise as the signal decays. However, the signal lengths used for field measurement were less and varied between 5 ms and 50 ms according to the targeted temporal resolution. The actual fitting durations are now stated in the manuscript. To avoid bias in the fitting results, residual field non-uniformity within the probe must be prevented from causing significant dephasing during the fitting duration. This is ensured by limiting the latter (up to 50 ms) to a fraction of the inverse of the frequency width (480 ms) and thus also to well less than the apparent T_2^* of 150 ms. A respective statement has been added to the text.

Lastly, in supplementary Figure 2, "to closely spaced sensors" should be "two closely spaced sensors."

Thank you spotting this glitch, which has been corrected.

Reviewer #2

The authors have carefully taken into account the input of all three referees and have substantially modified and improved the paper. The changes are also extensively described in the Rebuttal letter.

I think the paper should now be accepted for publication in its current form.

Reviewer #3

Referee Report of resubmitted version:

“Dynamic NMR field sensing with part-per-trillion resolution”

Gross, C. Barnett, G. Kervern, B.E. Dietrich, D.O. Brunner, T. Schmid, K. P. Pruessman“

In the revised version the authors have addressed some of my concerns and improved the readability of the manuscript.

Unfortunately the authors have left open a key question with respect to the true signal to noise ratio, which according to unnumbered equation (2) in line 34 of the manuscript determines the magnetic field measurement precision.

The claimed SNR = 14000 @ 50 kHz bandwidth is very suspect for the following reasons:

1. Thanks to the authors clarifications the electrical parameters and the operational conditions for the input circuit are now known. Given the inductivity of $L = 70 \text{ nH}$, quality factor $Q = 36$ @ $\omega = 2 \text{ Pi } 500 \text{ MHz}$ we can calculate the AC resistance R of the coil, which is $R = \omega L/Q = 6.1 \text{ Ohm}$. The impedance of the tuned (but not matched) circuit would be $R_t = R Q^2 = 7.9 \text{ kOhm}$, which without matching results in a Johnson noise of the purely tuned circuit of about $N_t \sim 11.4 \text{ nV/Hz}^{(-1/2)}$. This of course is not the noise seen at the input of the preamplifier. The tuned and matched input circuit has an impedance of 50 Ohm , which correspond to a Johnson noise of $N \sim 0.91 \text{ nV/Hz}^{(-1/2)}$. At 50 kHz measurement bandwidth this results in a minimum noise of $N(50 \text{ kHz}) = 0.9 \text{ nV/Hz}^{(-1/2)} (50000 \text{ Hz})^{(1/2)} = 204 \text{ nV} \sim 0.2 \text{ } \mu\text{V}$. The authors corrected my estimated NMR signal amplitude from $\text{EMF} \sim 200 \text{ } \mu\text{V}$ to $700 \text{ } \mu\text{V}$, which I believe now to be true. This means that the best possible SNR achievable at the input coil in the presented experiment is $\text{SNR}_{\text{max}} = 700 \text{ } \mu\text{V}/0.2 \text{ } \mu\text{V} = 3500$, and not 14000 as claimed. Even a perfect preamplifier with a noise figure of 0 dB would not improve SNR_{max} beyond this ultimate limit, and neither will do any postprocessing, digital filtering etc. for a single scan.
2. In reality, there will always be a noise contribution from the preamplifier (PMA 5452 with unknown input current and input voltage noise) and additional noise sources which in practice can only diminish the $\text{SNR}_{\text{max}} = 3500$ observed. The only way for a fair measurement of the SNR which is essential for the parts-per-trillion claim of the manuscript is to show the raw data of the FID and the corresponding base line fluctuations (expanded appropriately). Fig.2 should be shown in recorded Volts over time and SNR could be determined by taking the maximum amplitude (e.g. $V_{\text{peak}} - \text{peak}/2$) dividing by the RMS voltage value of the noise fluctuations.
The authors themselves admit that the baseline of the measurement “would however merely show a common stochastic noise trace”, but their claimed baseline for a $\text{SNR}=14000$ and 14 bit digital resolution cannot be a stochastic noise trace for it would only span the range of the least significant digital bit. For these two reasons, it is mandatory to show the reader the true baseline noise and the true SNR obtained in the experiments.

I think the authors should not destroy their remarkable results (Figs. 3, 4) by claiming a SNR which is not consistent with basic physics. Even if the claimed SNR was only a mere 1400, i.e. a magnitude lower than what was claimed, the resolution of the instrument would still be an impressive 10 parts-per-trillion and could be published in NatComm.

Again we greatly appreciate the reviewer's detailed consideration. Thank you very much. Prompted by the reviewer's input we have reconsidered our understanding of the sensitivity analysis in depth. After going through it in detail we conclude that the discrepancy between the two analyses is caused by different accounting for how noise and the EMF voltage are transformed in the resonant circuit. Please let us expand in the following:

As discussed by the referee, in typical NMR coil setups, the dominant source of noise is the solenoid coil. The AC resistance of the coil can be estimated from its quality-factor Q_{coil} . For a power-matched circuit

$$Q_{circuit} = \frac{1}{2} Q_{coil}.$$

$Q_{circuit}$ was measured to be 36. For an inductance of 70 nH and a resonance frequency of $\omega_0 = 2\pi \cdot 300$ MHz, the AC resistance of the coil is thus

$$R_{coil} = \omega_0 \frac{L}{Q_{coil}} = 1.83 \Omega$$

At room temperature, this corresponds to a Johnson-Nyquist RMS noise voltage at the coil terminals of

$$\sqrt{v_n^2} = \sqrt{4k_B T R_{coil}} = 0.17 \text{ nV}/\sqrt{\text{Hz}}.$$

On the other hand, the EMF induced in the coil was determined at

$$EMF_{max} = 700 \mu\text{V}$$

Both voltages, the EMF and the thermal noise voltage originate in the same physical structure, the coil. Therefore the tuning and matching circuitry transforms both voltages identically, i.e., their ratio is constant:

$$\frac{EMF_{max}}{\sqrt{v_n^2}} =: SNR \sqrt{BW} = 4.1 \cdot 10^6 \sqrt{\text{Hz}}$$

Additional SNR losses are introduced by the cable and the T/R-switch, connecting the probe to the preamplifier. Their total effect is on the order of 0.6 dB, corresponding to 7% in amplitude. The SNR at the preamplifier is therefore

$$SNR_{before\ preamp} \sqrt{BW} = 3.8 \cdot 10^6 \sqrt{\text{Hz}}$$

Considering the noise-figure of the preamplifier, which is another 0.6 dB, the SNR after the preamp is

$$SNR_{after\ preamp} \sqrt{BW} = 3.5 \cdot 10^6 \sqrt{Hz}$$

At a bandwidth of 50 kHz, this corresponds to a value of 15'600, corresponding well to the measured value of ~14'000.

Regarding the bit depth of the digitisation, we would like to point out that the analogue signals (after pre-amplification) are sampled at an ADC rate of 250 MHz. At this bandwidth, the SNR of the signals is only of the order of $\frac{3.5 \cdot 10^6 \sqrt{Hz}}{\sqrt{250 \cdot 10^6 Hz}} \approx 220$, for which a bit depth of 14 bits is well sufficient.

The large SNR gain when going from 250 MHz to 50 kHz bandwidth (SNR going from 220 to 15'600) arises from the elimination of all noise beyond 50 kHz by low-pass filtering prior to decimation according to common radio technology.

The figure below shows a dataset suitable for an SNR determination. It shows essentially the same FID signal as Fig. 2 but was complemented with a noise trace as requested by the reviewer. The noise signal was acquired immediately after the FID, using the same setup and spectrometer settings yet no excitation pulse. The normalized signals are proportional to the measured voltage at the ADC.

We hope that the comments above and this illustration settle the reviewer's concerns. If considered helpful, the figure could readily be added as supplementary material. We would like to leave this to the reviewer's and editor's discretion.

The rms noise voltage was 7.14e-5. The corresponding SNR is thus 14'006.

Reviewer 3:

The authors have given experimental evidence that their claimed SNR= 14000 at a bandwidth of 50 kHz. Their measured noise trace, showing a rms voltage of 7.14×10^{-5} relative to the normalized signal voltage of 1, supports the claimed SNR ratio. I strongly suggest to include the rms-voltage together with the signal into the supplementary together with the statement, that the red line in the signal curve is the noise trace without pulse excitation and that the shown insert corresponds to the red line scaled up by a factor of about 10000. Please include a another noise trace, showing also an expanded time axis ranging from 0 - 0.5 ms, i.e. at a time scale that makes individual data points visible and accessible to analysis.

However the reasons given by the authors for explaining SNR = 14000 are still mysterious for the following reasons:

1. The statement $Q_{\text{circuit}} = 0.5 Q_{\text{coil}}$ for a power matched circuit is only true if the loss in the capacitor is equal to the loss in the coil. In fact at 300 MHz we have $Q > 1000$ for high quality capacitors (e.g. Johanson, R14S series < 15 pF) with a few pF capacity, so most of the losses come from the coil and we get $Q_{\text{circuit}} = Q_{\text{coil}} = 36$ and $R_{\text{coil}} = 3.66$ Ohm and not 1.83 Ohm as stated by the authors.
2. The formula $R_{\text{coil}} = \omega_0 L / Q_{\text{coil}}$ is the ac impedance of a coil, not of a tuned circuit! At resonance the impedance of a purely tuned circuit (without matching) is given by $R_{\text{LC}} = R_{\text{coil}} Q_{\text{coil}}^2$, which is in the range of several kOhm -100 kOhm. The impedance of a tuned and matched LC circuit can be changed much below the kOhm range, typically to 50 Ohm in order to match with the input impedance of the preamplifier (typically 50 Ohm).
3. The authors explicitly state in their defense (page 5, paragraph 2) that they have a power matched circuit. In addition at page 13, line 241, and line 251 of the main text there is a clear statement that the LC circuit is matched to $Z_{\text{LC}} = 50$ Ohm = $Z_{\text{opt}} = 50$ Ohm of the preamplifier. In a standard view power match means that the best transfer of power from the NMR input circuit to the pre-amplifier is given by $Z_{\text{LC}} = Z_{\text{opt}}$. So we must assume the impedance of the input circuit as 50 Ohm (and not $R_{\text{coil}} = 1.83$ Ohm), and the associated input-noise is at least the corresponding Johnson noise (~ 0.9 nV/Hz^{1/2}), resulting at a given EMF of 700 μ V in SNR = 3500 at 50 kHz bandwidth. If the authors really tuned the input circuit to 1.83 Ohm then there is a large impedance mismatch with strong reflection of the signal. In that case about one third of the signal voltage is transferred from the LC circuit to the pre-amplifier. Again in this case the SNR would be much less than 14000.
4. If the authors made a noise match instead of a power match then they should tell the reader in detail how they accomplished these noise match and how they got a SNR =14000 at 50 kHz bandwidth including pre-amplifier voltage and corresponding current noise.
5. The $\text{EMF}_{\text{max}} = 700$ microV, given by the authors, is compared to the noise voltage to the untuned input coil alone (1.83 Ohm). If the EMF_{max} is measured in the tuned LC circuit, this voltage comparison is not valid for it compares a tuned to an untuned circuit.

In summary the data presented suggests a SNR = 14000 at 50 kHz bandwidth. However, in my opinion, the explanation cannot be followed through by the reader and I leave it to the editors to decide whether the manuscript is suitable for publication in its current state.

I would like the authors to explain their measured SNR again, and resubmit a new version with careful attention to the whole tuned and matched LC circuit, the calculation of EMF values in tuned LC circuits and the use of sampling rates versus bandwidth.

Again we would like to thank the referee for the in-depth considerations. We are glad that the added noise data is convincing as experimental evidence and we are happy to include the complementary material as requested. It will indeed be valuable corroborating an important point.

Regarding the plausibility of the SNR findings we have gone through the referee's comments (previous and current) in detail and compared them with our own view. In doing so we have identified one key respect in which the referee's and our perspectives appear to differ, namely the effect of impedance transformation on SNR. It is true that matching the detector coil (1.83 Ohm) to 50 Ohm increases the noise voltage correspondingly. However, it transforms the NMR signal voltage by the same ratio, leaving the SNR unaltered (except for a very slight loss due to losses in the matching capacitor). Therefore it is adequate to determine the SNR either from the plain coil's resistance and the EMF in the coil (similar to the calculations performed by Hoult and Richards in their SNR Paper: Hoult, D. I. & Richards, R. E. The signal-to-noise ratio of the nuclear magnetic resonance experiment. *J. Magn. Reson.* **24**, 71–85 (1976)) or from the 50 Ohm resistance and the *up-transformed* EMF. It is our impression that the large discrepancy in expected SNR between the referee's analysis and ours relates to the omission of the latter up-transformation (see also point 3 below).

To the specific points of critique we wish to respond as follows:

- 1) We would insist that the relation between the Q-factor of the matched circuit, commonly denoted as Q_{loaded} and the Q-factor of the unmatched circuit, $Q_{unloaded}$ is in fact

$$Q_{unloaded} = 2 Q_{loaded}$$

for NMR probes. This is widely accepted in the NMR community and derivations can be found in several textbooks, e.g., in Ching-nien Chen and David Hoult, *Biomedical Magnetic Resonance Technology*, Adam Hilger, Bristol and New York, 1989, page 169.

To still address the referee's concern without relying on said formula, we have re-determined the coil resistance with another method (described in the same book: Chen and Hoult, p. 153). It is based on measuring the Q-factor of the circuit both matched and unmatched. For the tuned and matched circuit we obtained $Q_{loaded} = 37$ as previously. We then unsoldered the matching capacitor (bringing the resonance frequency up to 313.5 MHz) to measure the Q-factor of the unloaded circuit. The measured value was $Q_{unloaded} = 68$, confirming the relation given above to within 10% and confirming the coil's AC resistance at

$$R_{coil}(313.5 \text{ MHz}) = \frac{\omega_0 L}{Q_{unloaded}} = 2 \Omega$$

for $L = 70 \text{ nH}$ as determined from the resonance frequency of the tuned circuit.

- 2) We agree with these statements but, frankly, are not sure in which respect our data or text conflict with them.
- 3) The circuit is indeed power-matched, i.e., the tuned coil is matched to the 50-Ohm impedance of the preamplifier by adding a capacitor. The matching acts as an impedance transformation, which transforms voltage along with impedance. It scales the EMF and thermal noise from the coil by the same factor such that their ratio, the SNR, remains unchanged. Particularly, the EMF of 700 μV induced in the coil should not be compared with 0.9 $\text{nV Hz}^{-1/2}$ of rms thermal noise at 50 Ohm (i.e., after transformation). It is rather to be compared with the noise level of

0.17 nV Hz^{-1/2} that corresponds to the AC resistance of the coil and which yields the stated maximum SNR of 4.1E6 Hz^{1/2}.

- 4) Noise matching would indeed be a different scenario. In this work, however, the probes are power-matched throughout.
- 5) Both the EMF and the thermal noise are calculated for an untuned coil. Again, tuning and matching change the voltages, but not their ratio.

Finally, the referee suggests explaining the SNR again and giving more attention to the tuned and matched LC circuit, the calculation of EMF, and differences between sampling rate and bandwidth. Regarding explanation of the SNR we hope that the remarks above have clarified why we believe that the observed SNR matches what is to be expected for the given coil impedance and EMF. Therefore we do not quite see a need to raise related questions in an expansion of the paper. Similarly, the other points listed above are clearly relevant background but bear, in our opinion, no unexpected insight.

(1) The authors wrote:

“Regarding the plausibility of the SNR findings we have gone through the referee’s comments (previous and current) in detail and compared them with our own view. In doing so we have identified one key respect in which the referee’s and our perspectives appear to differ, namely the effect of impedance transformation on SNR. It is true that **matching the detector coil (1.83 Ohm) to 50 Ohm** increases the noise voltage correspondingly. “

My view is that this misses the fact that the ‘coil’ is tuned, i.e. we have a resonant parallel LC circuit which has an impedance of $Z = R_{\text{coil}} * Q^2$ at resonance, where $R_{\text{coil}} = 1.83 \text{ Ohm}$ is the author’s AC resistance. With $Q=68$ (unloaded) we get 8.5 kOhm impedance. It is important to notice that this impedance is real and could equivalently be replaced by a 8.5 kOhm resistor. Both, the resistor and the LC circuit have a corresponding Johnson noise of $11.8 \text{ nV/Hz}^{1/2}$. Now the matching transforms this high impedance to 50 Ohm (power match) in order to maximize the signal transfer to the preamplifier which has 50 Ohm impedance. So we have downconversion from 8.5 kOhm to 50 Ohm, corresponding to a voltage noise of $0.91 \text{ nV/Hz}^{1/2}$ (and a correspondingly increased current noise).

“However, it transforms the NMR signal voltage by the same ratio, leaving the SNR unaltered (except for a very slight loss due to losses in the matching capacitor). Therefore it is adequate to determine the SNR either from the plain coil’s resistance and **the EMF in the coil** (similar to the calculations performed by Hoult and Richards in their SNR Paper: Hoult, D. I. & Richards, R. E. The signal-to-noise ratio of the nuclear magnetic resonance experiment. J. Magn. Reson. 24, 71–85 (1976)) or from the 50 Ohm resistance and the **up-transformed EMF**. It is our impression that the large discrepancy in expected SNR between the referee’s analysis and ours relates to the omission of the latter up-transformation (see also point 3 below).”

The statement that the SNR can be calculated from the plain coils resistance and the EMF in the coil is certainly true. I estimated the EMF to be 200 μV and the authors claimed 700 μV (which is ok with me). The crucial point is that this EMF is calculated and measured in a tuned circuit, where according to standard NMR theory the EMF is multiplied by the Q factor of the circuit and the noise gets multiplied by $Q^{1/2}$. So when calculating the SNR of the circuit you have to take the EMF of the circuit divided by the noise of the circuit and not only the noise of the untuned plain coil. You have just ignored this minute aspect of the calculation. I explicitly stated in my previous review (point 2) that “At resonance the impedance of a purely tuned circuit (without matching) is given by $R_{\text{LC}} = R_{\text{coil}} Q_{\text{coil}}^2$.”

To which you only responded:

“2) We agree with these statements but, frankly, are not sure in which respect our data or text conflict with them.”

That is where the crucial difference lies.

Again, your SNR cannot be 14.000.

(2) Concerning the presented base line in the Supplementary Figure 1:

I still wonder how the inset showing the noise on the $1e-4$ scale (100 μV) can be obtained with a 14 bit sampling system (stated on line 255 of the main text: “a high speed analogue-to-digital-converter (250 MS s^{-1} , **14 bit**).”

The voltage scale is normalized to 1, giving a minimum bit-step voltage of $1/2^{14} = 6.1e-5$ for a 14 bit sampling system. However, when looking at the vertical steps in the figure these seems to be resolved to at least 8 bit peak-peak in the inset shown, suggesting a $14+8 = 22$ bit sampling system at least.

The authors should explain in detail to the reader where this contradicting resolution comes from and whether the presented SNR of 14.000 is possibly an artefact from this exaggerated resolution. Again, for analysis of the noise, I would have preferred a detailed graph where each data point in the noise recording is resolved in the time and voltage domain of the original sampling straight from the ADC to minimize later interpretation errors and misleading conclusions regarding SNR.

It is possible that shown baseline graph has been obtained through a massive down-sampling from 250 MSamp/s to e.g. 50kSamp/s which could explain the fine vertical resolution. But this is the result of data processing and it is likely that you could produce any SNR desired if the sampling rate was only high enough. This possibly misleading point should be cleared up.

Since I believe this is an intriguing paper, I recommend publication after clearing up the points mentioned above.

Response to Review Comments

Please find our responses below. The reviewer's comments are shown in black. Our responses are interspersed in a blue font. Complementary details of potential interest are found in the subsequent appendix.

(1)

My view is that this misses the fact that the 'coil' is tuned, i.e. we have a resonant parallel LC circuit which has an impedance of $Z = R_{\text{coil}} * Q^2$ at resonance, where $R_{\text{coil}} = 1.83 \text{ Ohm}$ is the author's AC resistance. With $Q = 68$ (unloaded) we get 8.5 kOhm impedance. It is important to notice that this impedance is real and could equivalently be replaced by a 8.5 kOhm resistor. Both, the resistor and the LC circuit have a corresponding Johnson noise of $11.8 \text{ nV/Hz}^{1/2}$. Now the matching transforms this high impedance to 50 Ohm (power match) in order to maximize the signal transfer to the preamplifier which has 50 Ohm impedance. So we have downconversion from 8.5 kOhm to 50 Ohm, corresponding to a voltage noise of $0.91 \text{ nV/Hz}^{1/2}$ (and a correspondingly increased current noise).

We agree with this analysis. The tuning and matching steps do amount to up- and subsequent down-conversion, respectively.

The statement that the SNR can be calculated from the plain coils resistance and the EMF in the coil is certainly true. I estimated the EMF to be $200 \mu\text{V}$ and the authors claimed $700 \mu\text{V}$ (which is ok with me).

We believe that these statements establish common ground. The first sentence describes exactly how we arrive at our SNR estimate (following Hoult et al.): based on the plain coil's resistance and the EMF in the coil. We regret that we have not made this clear enough previously. To remove ambiguity, the respective passage in the Methods section (under *Electric properties of the sensor head*) now reads

"Its [the coil's] equivalent series resistance at 298 MHz was determined at 1.8Ω . The maximum electromotive force (EMF) induced in the coil by the precessing nuclear magnetization of a $^1\text{H}_2\text{O}$ sample was determined at $700 \mu\text{V}$ (40). In comparison, the coil resistance translates into thermal noise of $0.17 \text{ nV Hz}^{-1/2}$ at room temperature, resulting in an upper SNR bound of $\xi = 4.1 \times 10^6 \text{ Hz}^{1/2}$."

The crucial point is that this EMF is calculated and measured in a tuned circuit, where according to standard NMR theory the EMF is multiplied by the Q factor of the circuit and the noise gets multiplied by $Q^{1/2}$.

Here we do not agree with the premise. As stated in the preceding response we calculate the EMF (and the noise voltage) for the plain coil, not for the tuned coil. The reviewer expressed agreement with this approach ("The statement that the SNR can be calculated from the plain coils resistance and the EMF in the coil is certainly true.").

So when calculating the SNR of the circuit you have to take the EMF of the circuit divided by the noise of the circuit and not only the noise of the untuned plain coil. You have just ignored this minute aspect of the calculation.

We agree that one would have to work with the noise of the circuit when starting from the EMF of the circuit. However, this is not how we did the calculation (see preceding responses). Were one to calculate signal and noise at the whole-circuit (or tuned-coil) level then both signal and noise would

be scaled up, resulting in the same SNR estimate up to losses in the circuitry. To clarify this we have added a related passage in the Methods section under *Electric properties of the sensor head*.

I explicitly stated in my previous review (point 2) that

“At resonance the impedance of a purely tuned circuit (without matching) is given by $R_{LC} = R_{coil} / Q_{coil}^2$.”

To which you only responded:

“2) We agree with these statements but, frankly, are not sure in which respect our data or text conflict with them.”

That is where the crucial difference lies.
Again, your SNR cannot be 14.000.

We hope that this controversy is removed upon agreement on the preceding points. The circuit consideration does not apply if we agree on the validity of estimating the SNR at the plain coil level. We have no doubt that the reported SNR is real. It is not only in line with the theoretical estimate but also the result of measurement as the reviewer previously acknowledged.

(2) Concerning the presented base line in the Supplementary Figure 1:

I still wonder how the inset showing the noise on the $1e-4$ scale ($100\mu V$) can be obtained with a 14 bit sampling system (stated on line 255 of the main text: “a high speed analogue-to-digital-converter (250 MS s⁻¹, 14 bit).”

As the reviewer suspects later in this comment and as we stated in the manuscript our receiver system does include digital band-pass filtering and down-sampling in addition to mere digitization. This is a standard approach in digital radio receivers for limited-band signals. The purpose of the filtering steps is to remove out-of-band noise and thus increase the time-domain SNR. The SNR increase requires an increase also in the number of bits used to represent the signal. In our system the number of bits increases from 14 bit at the ADC output to 32 bit at the eventual output (for each its real and imaginary part).

For the benefit of the reader we have added a passage explaining these considerations in the Methods section (under *Radiofrequency spectrometer*).

The voltage scale is normalized to 1, giving a minimum bit-step voltage of $1/2^{14} = 6.1e-5$ for a 14 bit sampling system. However, when looking at the vertical steps in the figure these seems to be resolved to at least 8 bit peak-peak in the inset shown, suggesting a $14+8 = 22$ bit sampling system at least.

This is correct. The SNR increase relative to the ADC output, according to the bandwidth reduction, is $\left(\frac{50 \text{ kS s}^{-1}}{250 \text{ MS s}^{-1}}\right)^{1/2} = 71$, which corresponds to a need for additional 7 bits of resolution.

The authors should explain in detail to the reader where this contradicting resolution comes from and whether the presented SNR of 14.000 is possibly an artefact from this exaggerated resolution.

A passage for this purpose has been added to the Methods section (under *Radiofrequency spectrometer*).

Again, for analysis of the noise, I would have preferred a detailed graph where each data point in the noise recording is resolved in the time and voltage domain of the original sampling straight from the ADC to minimize later interpretation errors and misleading conclusions regarding SNR.

This request is, unfortunately, not practical for us to implement. The ADC is directly connected to an FPGA for digital processing and not designed to enable storage of raw data. Limits of flexibility of the hardware relate to the fact that digital signals at 250 million samples per second (3.5 Gb/s) are a complex matter in themselves. At this bandwidth proper data transmission is a radiofrequency problem and requires highly specified interfaces to downstream modules. The only viable option for recording ADC outputs would be to leave the FPGAs in place and redevelop their hardware description language (HDL) configuration. However, this would be a major undertaking. The only point that it would illustrate is the relationship between bandwidth and SNR, which is truly well known. Notably, the primary SNR metric, ξ , that we use for both the SNR estimate and for reporting measured SNR, is even independent of bandwidth (having units of $\text{Hz}^{1/2}$). So it will be the same at the ADC and system outputs. On this basis, frankly, we consider the effort of reconfiguring the system out of proportion to the insight to be gained.

It is possible that shown baseline graph has been obtained through a massive down-sampling from 250 MSamp/s to e.g. 50kSamp/s which could explain the fine vertical resolution. But this is the result of data processing and it is likely that you could produce any SNR desired if the sampling rate was only high enough. This possibly misleading point should be cleared up.

The reviewer is perfectly right: the sampling rate of the shown data is 50 kS/s, down-sampled from the ADC output of 250 MS/s. Both numbers and the fact that down-sampling is part of the processing were mentioned in the original manuscript but we acknowledge that this may not have been clear enough. We have therefore expanded the respective passage in the Methods section (under *Radiofrequency spectrometer*).

The SNR could not be increased by faster initial sampling since faster sampling would take in more noise power in proportion to the bandwidth increase. The bandwidth-corrected SNR ξ would remain unchanged.

Since I believe this is an intriguing paper, I recommend publication after clearing up the points mentioned above.

Appendix

As agreed, the SNR of the NMR experiment can be calculated either from the plain coil's EMF and the plain coil's resistance or the EMF and resistance of the tuned circuit.

Our approach has been to calculate the SNR from the *plain* coil for simplicity. We did so as described in Hoult, D.I., Richards, R. E. The Signal-to-Noise Ratio of the Nuclear Magnetic Resonance Experiment. *J. Magn. Reson.* **24**, 71-85 (1976). For our coil specs, we obtained $4 \times 10^6 \text{ Hz}^{1/2}$, in good agreement with the measurements (see manuscript Fig. 2 and previous rebuttal letters).

For these calculations an EMF of 700 μV was used, i.e., the EMF induced in the *plain* coil, not the resulting voltage in the *tuned* coil. The EMF was calculated using the method described by Hoult and Richards, fully accounting for the coil geometry and magnetisation distribution after NMR excitation. The order of magnitude of this result can be cross-validated using Faraday's law:

Coil area	$A \approx 4 \times 10^{-6} \text{ m}^2$
Magnetisation	$M = 4 \times 10^{-9} \cdot 7 \text{ T}$
Frequency	$\omega = 300 \times 10^6 \cdot 2\pi \text{ s}^{-1}$

$$EMF = - \frac{d\Phi}{dt} \approx \omega AM = 200 \mu\text{V per coil turn}$$

Alternatively, in a standard textbook (A. Abragam, *Principles of Nuclear Magnetism*, Oxford Science Publications, 1961), Abragam provides a formalism for calculating the SNR in a *tuned* coil (eq. 76, p. 83, in the 2006 reprint). Inserting our coil specs we obtain an SNR estimate of the order of $4 \times 10^6 \text{ Hz}^{1/2}$ also for the *tuned* coil. The equivalency of the two formalisms was formally proven by Hoult and Richards in the appendix to their paper.